# Electrostatics from Laplacian Eigenbasis for Neural Network Interatomic Potentials

## Abstract

In this work, we introduce $\Phi$-Module, a universal plugin module that enforces Poisson's equation within the message-passing framework to learn electrostatic interactions in a self-supervised manner. Specifically, each atom-wise representation is encouraged to satisfy a discretized Poisson's equation, making it possible to acquire a potential $\phi$ and a corresponding charges $\rho$ linked to the learnable Laplacian eigenbasis coefficients of a given molecular graph. We then derive an electrostatic energy term, crucial for improved total energy predictions. This approach integrates seamlessly into any existing neural potential with insignificant computational overhead. Our results underscore how embedding a first-principles constraint in neural interatomic potentials can significantly improve performance while remaining hyperparameter-friendly, memory-efficient and lightweight in training.

## 1 Introduction

In quantum chemistry, the task of correct prediction of atomic energies is paramount, but stands a great challenge due to extensive computational requirements of *ab-initio* methods like Density Functional Theory (DFT) (Hohenberg & Kohn, 1964; Kohn & Sham, 1965). Modern deep learning presents a way to solve this problem with geometric graph neural networks (GNN). GNNs operate on molecular graphs by exchanging messages between nodes and edges, learning meaningful representations in the process. In recent years, a series of molecular modeling methods have been developed (Gasteiger et al., 2021; Wang et al., 2022; Passaro & Zitnick, 2023; Musaelian et al., 2023).

While those models and their alternatives demonstrate competitive performance, they rely on message passing which is local in nature (Dwivedi et al., 2022). The issue arises as molecular interactions are described using both local and non-local interatomic interactions. Local interactions include bond stretching, bending and torsional twists. They can be easily captured by message propagation in modern GNNs for molecular graphs (Zhang et al., 2023). At the same time, non-local interactions like electrostatics or van der Waals forces can span long distances and have cumulative effect (Stone, 2013). The main drawbacks of representations learned by GNNs are over-smoothing (Rusch et al., 2023) and over-squashing (Alon & Yahav, 2020) interfere the precise modeling of non-local interactions.

To tackle the problem of learning non-local interactions, a number of customizations have been proposed for molecular GNNs. Some of those require prior data in the form of partial charges or dipole moments, which is costly to retrieve using DFT (Unke & Meuwly, 2019; Ko et al., 2021), or carry inaccurate information derived from pre-defined empirical rules (Gasteiger & Marsili, 1978). Another distinct direction of research proposes merging of message passing and Ewald summation (Ewald, 1921) to approximate electrostatic interactions (Kosmala et al., 2023; Cheng, 2024).

In this paper, we explore a new viewpoint on the problem of learning non-local atomistic and molecular interactions. Our aim is to learn electrostatic energy in a completely self-supervised manner without any external labeled data. To fulfill this goal, we propose $\Phi$-Module, a universal augmentation module, which can be embedded into any GNN. $\Phi$-Module relies on the connection between the Laplacian of a molecular graph and partial charges to improve the quality of the neural network interatomic potentials.

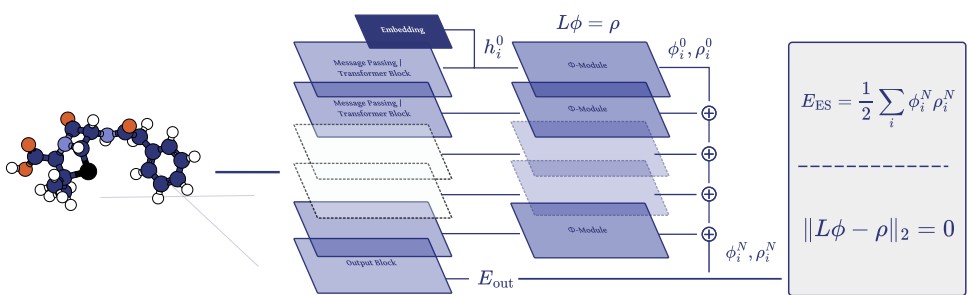

Figure 1: Overview of the proposed $\Phi$-Module. $\Phi$-Module encodes electrostatic constraints based on Poisson's equation into hidden representations of any neural network interatomic potential. $\Phi$-Module is integrated at each step of message passing. It uses lightweight convolutional submodule which we refer to as $\alpha$-Net to estimate coefficients of Laplacian eigenbasis directly from constantly updated atomic representations. Those eigenbasis coefficients are then used to optimize Poisson's equation residual $\|\mathbf{L}\phi - \rho\|_2 = 0$ and compute electrostatic energy term $\mathbf{E}^{\text{ES}}$ making an important contribution to predictions and leading to improved performance on computational chemistry problems. See Section 3.

Our contributions are highlighted as follows:

- We propose $\Phi$-Module, a plugin module for GNNs on molecular graphs, which learns electrostatic information from atomic embeddings with estimation of Laplacian eigenbasis coefficients. See Section 3.

- We demonstrate how $\Phi$-Module improves wide variety of baselines on challenging OE62 and MD22 benchmarks for energy prediction and molecular dynamics respectively. On OE62 addition of $\Phi$-Module results in error reductions from 5%. For MD22, the proposed solution improves baseline, which achieves best results among other models in 5 out of 14 cases and improves the baseline in 12 out of 14. See Section 4

- We provide valuable insights on the appealing properties of $\Phi$-Module. Namely, its hyperparameter stability, physically informative formulation, stability under data scarsity and memory-efficiency crucial to molecular modeling. See Section 4

## 2 BACKGROUND

**Message Passing Neural Network Potentials.** Geometric graph neural networks (GNN) function on molecular graphs with atoms as nodes $V \in \{\mathbf{x}_1, \ldots, \mathbf{x}_N\}$ and atomic bonds as edges $E \in \{(i,j) \mid i \neq j\}$. Each node and edge may include additional features $\mathbf{z}_i \in \mathbb{R}^{d_x}$ and $\mathbf{e}_{ij} \in \mathbb{R}^{d_e}$, which commonly are nuclear charge numbers and distances between nodes. The edges are constructed as a radius graph with a specific cutoff radius $r_c$ as a hyperparameter, such that $i \in \mathcal{N}(j)$ if $\|\mathbf{x}_j - \mathbf{x}_i\|_2 \leq r_c$. GNN initially encodes atoms solely on the basis of local properties producing $\mathbf{h}^0 \in \mathbb{R}^F$ features.

In the following steps, GNN refines the initial node representations by applying several iterations of message $\mathbf{m}_i^{(l)}$ aggregations and updates:

$$\mathbf{m}_i^{(l)} = \bigoplus_{j \in \mathcal{N}(i)} \left( \mathcal{M}^l \left( \mathbf{h}_i^{(l)}, \mathbf{h}_j^{(l)}, \mathbf{e}_{ij} \right) \right)$$

$$\mathbf{h}_i^{(l+1)} = \mathcal{U}^l \left( \mathbf{h}_i^{(l)}, \mathbf{m}_i^{(l)} \right),$$

where $\mathcal{M}^l$ is a learnable function, which constructs the message, $\mathcal{U}^l$ is another learnable function to update the representations of nodes with aggregated messages and $\bigoplus$ is the message aggregation operator, which is typically *sum* or *mean*. Finally, the resulting representations are processed to output the energy $\hat{\mathbf{E}}$. Neural network potentials are commonly optimized to approximate target

energy $\mathbf{E}$ of a given structure using L1 loss:

$$\mathcal{L}_{\text{model}} = \frac{1}{N} \sum_{i=1}^{N} \left| \mathbf{E}_i - \hat{\mathbf{E}}_i \right|.$$

**Poisson's Equation for Electrostatics.** Electrostatic interactions contribute a significant component of molecular energy, but they are not directly encoded in $\mathbf{E}^{\text{model}}$. To incorporate them, we begin from the classical definition of electrostatic energy in terms of charges $\boldsymbol{\rho}$ and potential $\boldsymbol{\phi}$:

$$\mathbf{E}^{\text{ES}} = \tfrac{1}{2} \sum_i \boldsymbol{\rho}_i \boldsymbol{\phi}_i. \tag{1}$$

This expression reflects the work required to assemble the system of charges under their mutual Coulomb interactions. The potential $\phi$ characterizes how a unit charge at node $i$ is influenced by all other charges in the system, and thus mediates central phenomena such as bond formation, molecular geometry stabilization, and long-range protein–ligand recognition.

In the continuous setting, $\phi$ and $\rho$ are linked through Poisson's equation $\nabla^2 \phi = -\rho/\varepsilon$, where $\nabla^2$ denotes the Laplace operator. For molecular graphs, the continuous Laplacian is naturally replaced by the graph Laplacian $\mathbf{L}$, which arises as a finite-difference approximation of the continuous operator on a discretized domain (Smola & Kondor, 2003). This yields the discrete Poisson equation

$$\mathbf{L}\boldsymbol{\phi} = \boldsymbol{\rho}. \tag{2}$$

Here, $\mathbf{L}$ captures local connectivity and encodes how the potential at each atom deviates from the average of its neighbors, thus mirroring the curvature-based interpretation of the Laplacian in Euclidean space.

The proposed $\Phi$-Module is designed to operate directly on Equation (2), enabling the model to learn consistent representations of $\phi$ and $\rho$ from atomic messages. In doing so, it approximates the electrostatic potential field on the molecular graph and provides the corresponding contribution $\mathbf{E}^{\text{ES}}$ to the total energy. This formulation tightly integrates graph-theoretic structure with physical inductive bias, bridging the gap between molecular electrostatics and message-passing neural architectures.

## 3 $\Phi$-MODULE

In this section, we describe in detail how to encode electrostatic constraints coming from Poisson's equation into representations learned by neural network potentials. Firstly, we explain the importance of learning $\phi$ and $\rho$ in the eigenbasis of $\mathbf{L}$. Next, $\alpha$-Net is introduced to learn the spectral coefficients essential to obtain the solution of the equation. Finally, we theoretically prove the appealing properties of the spectral decomposition approach in comparison with the direct learning of Poisson's equation components.

**Spectral Decomposition of Laplacian.** To infuse physical knowledge, resulting in improved learning dynamics, we propose to derive potential $\phi$ and charges $\rho$ in an eigenbasis of Laplacian $\mathbf{L}$. Note that $\mathbf{L}$ is identical for different 3D compositions of the same molecule, therefore we weigh Laplacian values by interatomic instances $\mathbf{d}_{ij} = \|\mathbf{x}_j - \mathbf{x}_i\|_2$ to be able to differentiate between molecular conformations.

Recall that $\mathbf{L}$ in Equation (2) is symmetric positive-semidefinite. Therefore, it can be decomposed as $\mathbf{L} = U\Lambda U^\top$, where $U = [\mathbf{u}_1, \dots, \mathbf{u}_n]$ are orthonormal eigenvectors and $\Lambda = diag(\lambda_1, \dots, \lambda_n)$ with $\lambda_1 \geq \lambda_2 \geq \cdots \geq \lambda_n \geq 0$ is the diagonal matrix of eigenvalues of $\mathbf{L}$.

Any vector $v \in \mathbb{R}^N$ can be expanded in the basis of $U$ as $v = U\boldsymbol{\alpha}$, where $\boldsymbol{\alpha}$ is the eigenbasis coefficients of $\mathbf{L}$. We expand the Poisson's Equation with two distinct vectors $\boldsymbol{\alpha}_\phi$ and $\boldsymbol{\alpha}_\rho$ using the spectral decomposition of $\mathbf{L}$ given potential and charge eigenbasis projections as

$$\phi = U\boldsymbol{\alpha}_\phi \tag{3}$$
$$\rho = U\Lambda\boldsymbol{\alpha}_\rho. \tag{4}$$

The exact formulation enables symmetric gradients of the residual, making optimization stable. Details on the theoretical difference between the two options are discussed in Appendix F.

**Self-Supervised Learning of Potential and Charges.** Equations 3 and 4 highlight that we need to estimate eigenbasis coefficients $\boldsymbol{\alpha}$ to calculate Poisson's equation residual. We propose learning $\boldsymbol{\alpha}_\phi$ and $\boldsymbol{\alpha}_\rho$ from node representations $\mathbf{h}$ using convolutional subnetwork called $\boldsymbol{\alpha}$-Net denoted $\boldsymbol{\alpha}_\theta$. $\boldsymbol{\alpha}$-Net (Figure 2) consists of a pair of 1D convolutional layers combined with global pooling to map node embeddings to a distinct number of eigenvalues to have them processed by two separate heads.

This lightweight architecture consistently computes the eigenbasis coefficients to update $\phi$ and $\rho$ at each iteration of the message passing. The specific design allows us to operate on hidden dimensions of $\mathbf{h}$ to compress the most essential information into a low-dimensional representation. We calculate $\phi$ and $\rho$ using Equation (3) and Equation (4) and $\boldsymbol{\alpha}$ coefficients obtained from $\boldsymbol{\alpha}$-Net sepaterely for $\phi$ and $\rho$. In the initial step of message passing, $\boldsymbol{\alpha}$ is computed from the node features after first message passing step $\mathbf{h}^1$. In the subsequent steps potential and charges are aggregated via summation operation as $\phi^N = \phi^{N-1} + U\boldsymbol{\alpha}_\theta(\mathbf{h}^N)$ and $\rho^N = \rho^{N-1} + U\Lambda\boldsymbol{\alpha}_\theta(\mathbf{h}^N)$.

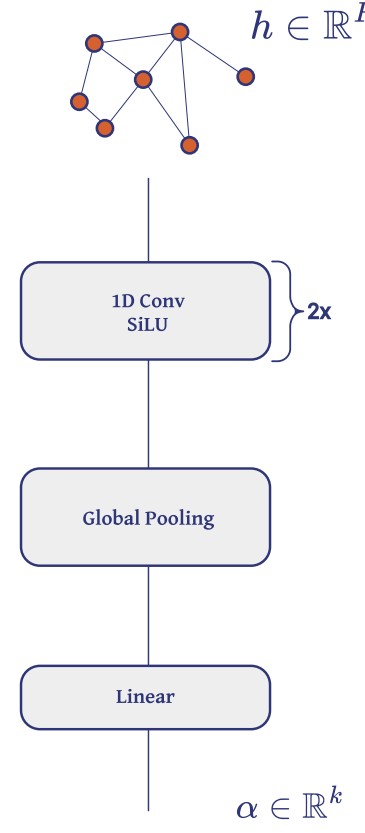

$h \in \mathbb{R}^F$

1D Conv
SiLU

2x

Global Pooling

Linear

$\alpha \in \mathbb{R}^k$

We compute residual $\mathcal{L}_{\text{PDE}} = \beta\|\mathbf{L}\phi - \boldsymbol{\rho}\|_2$ in order for $\phi$ and $\boldsymbol{\rho}$ to satisfy Equation (2), where $\beta$ is a hyperparameter controlling the impact of the $\Phi$-Module. If a training dataset consists of neutral molecules, we apply an additional constraint to enforce net zero charge $\mathcal{L}_{\text{net}} = \gamma|\sum_i \boldsymbol{\rho}_i|$, where $\gamma$ is a hyperparameter. The final training objective for energy prediction is $\mathcal{L} = \mathcal{L}_{\text{model}} + \beta\mathcal{L}_{\text{PDE}} + \gamma\mathcal{L}_{\text{net}}$.

**Electrostatic Energy.** After final message passing step we calculate electrostatic term $\mathbf{E}^{\text{ES}}$ using Equation (1). The complete energy is obtained as a combination of energy $\mathbf{E}^{\text{model}}$ retrieved from the model and electrostatic term as $\hat{\mathbf{E}} = \mathbf{E}^{\text{model}} + \mathbf{E}^{\text{ES}}$.

**Theoretical Justification.** We inspect theoretical properties of $\Phi$-Module below. In Theorem 3.1 we demonstrate the strict convexity of the optimization problem of $\rho$. Following this results, we prove monotone improvement relative to the error in Theorem 3.2. Proofs can be examined in Appendix H.

**Theorem 3.1** (Exact inner minimizer over $\rho$). *Define $a = \mathrm{E} - \mathrm{E}_{model}$. Fix $\phi \in \mathrm{span}(U_k)$. The unique minimizer of $\rho \mapsto \mathcal{L}(\phi, \rho)$ over $\mathrm{span}(U_k)$ is*

$$\rho^\star(\phi) = L\phi - t^\star(\phi)\,\phi, \qquad t^\star(\phi) = \frac{a + \frac{1}{2}\phi^\top L\phi}{2\beta + \frac{1}{2}\|\phi\|^2}.$$

**Theorem 3.2** (Monotone objective decrease in optimization towards $\rho^\star$). *Define $A(\phi) := a + \frac{1}{2}\phi^\top L\phi$. Then substituting $\rho^\star(\phi)$ from Theorem 3.1 yields*

$$\widetilde{\mathcal{L}}(\phi) := \mathcal{L}\big(\phi, \rho^\star(\phi)\big) = A(\phi)^2\,\frac{4\beta}{4\beta + \|\phi\|^2} \le A(\phi)^2,$$

*with equality if and only if $A(\phi) = 0$ or $\phi = 0$.*

Figure 2: $\alpha$-Net. It transforms dense atomic representations into sparser coefficients of Laplacian's eigenbasis to acquire potentials and charges. See Section 3.

**Intuition.** Theorem 3.1 tells us that optimization steps towards $\rho^\star$ are aligned with the potential field avoiding the usage of arbitrary modes. This couples changes in important frequency range to residual minimization. The true solution is recovered when the error is zero. Next, Theorem 3.2 shows provable improvements in the main objective resulting from optimization of $\Phi$-Module. Additionally, factor denominator acts as a damping on the energy term given $\|\phi\|^2$ is large.

**Implementation Details.** We implement spectral decomposition of $\mathbf{L}$ using the "Locally Optimal Block Preconditioned Conjugate Gradient" method (LOBPCG) (Knyazev, 2001). LOBPCG

enables us to compute only a selected amount of eigenvalues and gives the opportunity to process large macromolecules with the Φ-Module. This decision also keeps us away from the ambiguity of invariance and sorting of eigenvalues and eigenvectors during their computations - we strictly get $k$ selected eigenvalues and their corresponding eigenvectors without the need to sort them anyhow. Block-diagonal nature of $\mathbf{L}$ and independence of its blocks allow us to compute eigendecomposition once for a single batch in an efficient vectorized manner without relying on any paddings.

The pseudocode for integration of the Φ-Module can be seen in Section B. The proposed augmentation fits into any neural network that iteratively operates on atomic representations.

## 4 EXPERIMENTS

In this section, we conduct diverse experiments to establish the importance of Φ-Module. Firstly, performance of networks injected with the proposed module are tested against corresponding baselines on popular quantum chemical datasets and molecular dynamics. Next, we demonstrate that the Φ-Module exhibits robust hyperparameter stability, requiring minimal tuning to achieve improved performance. Additionally, we show clear benefits from the memory-scaling dynamics of the Φ-Module and provide evidence that current architectural choices encode physically meaningful priors. Finally, we evaluate the model in data-scarce regimes and show that it outperforms baselines even with limited supervision.

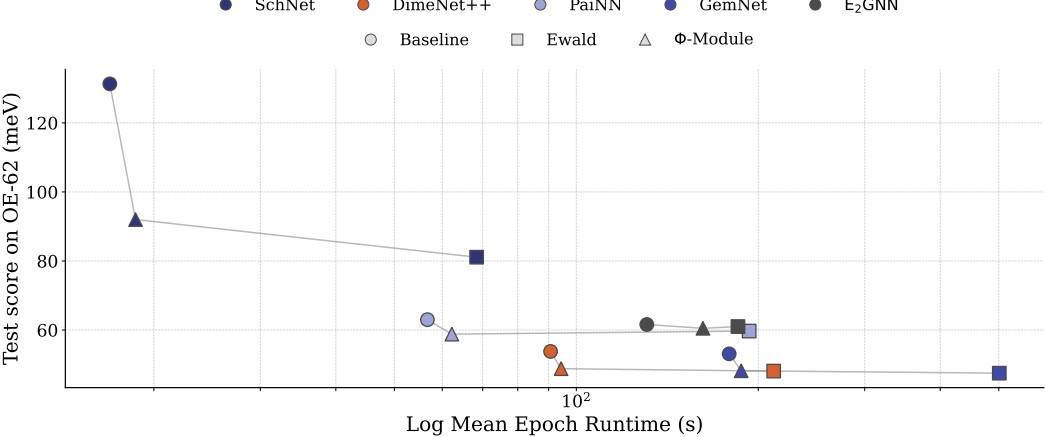

Figure 3: Energy MAEs and computation time of baselines and their alternatives with Φ-Module on OE62. Φ-Module achieves comparable error to Ewald summation at the same time being almost as fast as clean baseline. See Section 4.

In the following experiments, we modify various baseline models by integrating the Φ-Module and denote the resulting models with the prefix "Φ-". The Φ-Module introduces four tunable hyperparameters: $k$ — the number of eigenvalues; $\beta$ — the weight of the PDE loss $\mathcal{L}_{\text{PDE}}$ and $\gamma$ — the weight of the charge neutrality loss $\mathcal{L}_{\text{net}}$. For a detailed overview of the hyperparameter configurations, please refer to Section D.

**OE62.** We start our analysis with the challenging OE62 (Stuke et al., 2020) dataset to demonstrate how Φ-Module enhances neural network interatomic potentials. OE62 features about 62,000 large organic molecules with the energies calculated using Density Functional Theory (DFT). The molecules within OE62 have around 41 atoms on average and may exceed the size of 20 Å. Dataset is divided into training, validation and testing parts and preprocessed according to the previous studies (Kosmala et al., 2023).

Common baselines namely SchNet (Schütt et al., 2017), DimeNet++ (Gasteiger et al., 2020), PaiNN (Schütt et al., 2021), GemNet-T (Gasteiger et al., 2021) and $E_2$GNN (Yang et al., 2025) are trained on OE62. Their counterparts with Φ-Module are named accordingly as Φ-SchNet, Φ-DimeNet++, Φ-PaiNN, Φ-GemNet-T and Φ-$E_2$GNN. Φ-Module is compared against the baselines and models

with the Ewald message passing block (Kosmala et al., 2023). The computational cost is computed as the average time for one epoch given selected hyperparameters in Appendix D.

The results in Figure 3 demonstrate that Φ-Module improves performance for each baseline by a distinct margin ($\geq$ 5%) and for around 3% for $E_2$GNN outperforming the Ewald block in 2 out 5 cases with an evidently smaller computational overhead. The exact results can be found in Table 3 in Appendix E.

**MD22.** The MD22 (Chmiela et al., 2019) dataset is a comprehensive collection of molecular dynamics (MD) trajectories of biomolecules and supramolecules. It covers a wide range of molecular sizes, with atom counts spanning from 42 to 370 atoms per system. Each dataset represents a single molecule's dynamic behavior, comprising between 5,032 and 85,109 structural snapshots captured over time. The MD22 is split into training, validation and testing sets according to sGDML (Chmiela et al., 2019).

We benchmark the ViSNet (Wang et al., 2022) model and its Φ-ViSNet modification on seven presented molecule in Table 1 and demonstrate that our method achieves consistent improvements over the baseline in both energy and force predictions for most of the cases.

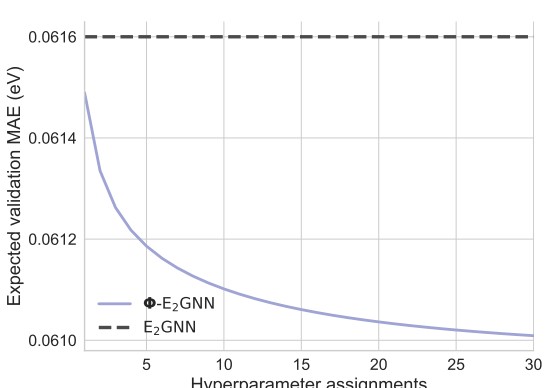

Figure 4: Test expected validation MAE for Φ-$E_2$GNN against the baseline model on OE62. Any choice of selected hyperparameters leads to improved performance, underlining tuning stability of the Φ-Module. See Section 4.

Original ViSNet achieves the best results only in 2 out of 14 cases, while Φ-ViSNet sets the best results for the 5 metrics of the measured setups. Additionally, Φ-ViSNet outperforms basic ViSNet in 11 out of 14 cases. The average computational overhead for the insertion of the Φ-Module is only 9%. Note, that no hyperparameter search was performed for MD22 due to limited available resources, hence the results may be improved in practice.

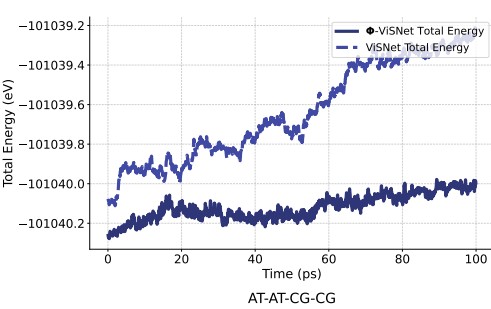

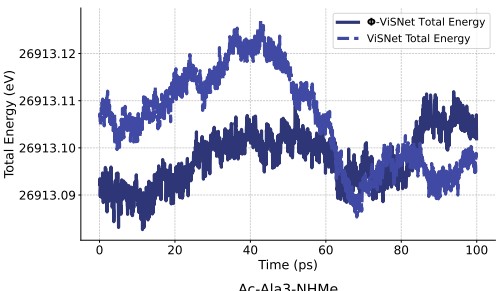

Figure 5: Total energy over 100 ps NVE simulation for (Left) AT-AT-CG-CG and (Right) Ac-Ala3-NHMe molecules obtained from baseline ViSNet and Φ-ViSNet. Energy drift is bounded at 0.0001% over the full trajectory for Φ-ViSNet in both cases. Moreover, it attains x4 and x2 smaller total magnitude of energy drift respectively compared to the baseline model. See Section 4.

**Stability of Molecular Dynamics Simulations** One of the crucial quantities of neural network interatomic potentials is the ability to show stable molecular dynamics simulation trajectories. It is regarded that low force errors do not directly guarantee stable simulations (Fu et al., 2022). For this purpose we conduct long molecular dynamics simulation for large molecules from MD22 with Φ-ViSNet. To demonstrate stability, we choose AT-AT-CG-CG and Ac-Ala3-NHMe. We perform microcanonical ensemble (NVE) simulation of 100 picoseconds (ps) duration with the Verlet integrator. One step is 0.5 femtoseconds (fs).

Table 1: Mean absolute errors (MAE) of energy (kcal/mol) and forces (kcal/mol/Å) for seven large molecules on MD22. The best one in each category is highlighted in bold, the second best is underlined. The runs where $\Phi$-ViSNet outperforms baseline are also underlined. See Section 4.

| Molecule | Type | Diameter | sGDML | SO3KRATES | Allegro | Equiformer | MACE | ViSNet | $\Phi$-ViSNet |
|---|---|---|---|---|---|---|---|---|---|
| Ac-Ala3-NHMe | Energy | 10.75 | 0.390 | 0.337 | 0.102 | 0.083 | **0.062** | 0.102 | 0.091 |
| | Forces | | 0.797 | 0.244 | 0.107 | **0.080** | 0.088 | 0.086 | 0.082 |
| DHA | Energy | 14.58 | 1.312 | 0.379 | 0.115 | 0.179 | 0.132 | 0.072 | **0.010** |
| | Forces | | 0.747 | 0.242 | 0.073 | **0.051** | 0.065 | 0.099 | 0.075 |
| Stachyose | Energy | 13.87 | 4.050 | 0.442 | 0.249 | 0.140 | 0.124 | **0.017** | 0.040 |
| | Forces | | 0.674 | 0.435 | 0.097 | 0.064 | 0.088 | 0.107 | **0.011** |
| AT-AT | Energy | 17.63 | 0.724 | 0.178 | 0.143 | 0.131 | 0.109 | **0.008** | **0.008** |
| | Forces | | 0.691 | 0.216 | 0.095 | 0.096 | 0.099 | **0.086** | 0.111 |
| AT-AT-CG-CG | Energy | 21.29 | 1.389 | 0.345 | 0.393 | 0.151 | 0.158 | 0.149 | **0.074** |
| | Forces | | 0.703 | 0.332 | 0.128 | 0.125 | **0.115** | 0.199 | 0.182 |
| Buckyball catcher | Energy | 15.89 | 1.196 | **0.381** | 0.526 | 0.398 | 0.481 | 0.937 | 0.741 |
| | Forces | | 0.682 | 0.237 | 0.089 | 0.111 | **0.085** | 0.690 | 0.631 |
| Double-walled nanotube | Energy | 32.39 | 4.012 | 0.993 | 2.210 | 1.195 | 1.655 | 1.023 | **0.506** |
| | Forces | | 0.523 | 0.727 | 0.343 | **0.275** | 0.396 | 0.680 | 0.593 |

We aim to achieve minimal energy drift as NVE's total energy remains constant up to small numerical fluctuations (see background in Appendix G). In this sense we can expose any non-conservative force field behavior. In Figure 5, total energy drift is bounded at 0.0001% relative to the baseline energy, which indicates that $\Phi$-Module can be used for stable long-range molecular dynamics simulations.

**Hyperparameter Stability.** In this section, we study the hyperparameter stability of the $\Phi$-Module. We employ *Expected Validation Performance* (EVP) (Dodge et al., 2019) which measures how performance of $\Phi$-E$_2$GNN trained on OE62 changes with the increasing number of hyperparameter assignments.

The hyperparameter search includes $k, \beta$. In Figure 10 EVP curve for $\Phi$-E$_2$GNN is below E$_2$GNN baseline performance line after hyperparameter search. The plot demonstrates that any configuration of hyperparameters results in improved performance against the baseline. This highlights the practical convenience of $\Phi$-Module in terms of hyperparameter choice. Detailed information on EVP and plots for other models can be examined in Section G.

**$\Phi$-Module Memory Scaling in Comparison with Ewald Summation.** To access one of the crucial benefits of the $\Phi$-Module - *memory efficiency*, we set up experiment to run SchNet, $\Phi$-SchNet, SchNet with Ewald message passing block (Kosmala et al., 2023), and SchNet with Neural P3M block (Cheng, 2024) on linear carbyne chains (Liu et al., 2013) of variable sizes from $10^3$ to $10^5$ atoms. We measure CUDA memory consumption on an NVIDIA 80GB H100 GPU in MBs.

The results can be seen in Figure 7. Ewald and Neural P3M quickly result in out-of-memory (OOM) error and do not scale favorably to large systems. This is an essential problem as electrostatic interactions die off much slower than other forms of long-range forces and are more evident in large systems. On the other hand, $\Phi$-Module demonstrates the same scaling as the baseline model showcasing its potential for extremely large molecules.

**Design Choices.** In this section, we elaborate on the main design choices made for $\Phi$-Module. We take models used for OE62 and train them while gradually disabling main parts of the proposed method. Firstly, we replace **L** with a random matrix to eliminate any physical grounding. Secondly, we remove the optimization of the residual of Poisson's equation as in Section 3 to test if unconstrained addition of trainable parameters is helpful. In the Figure 8, a distinct trend can be seen of the complete solution for $\Phi$-Module outperforming the version lacking physical grounding. This experiment supports the formulation proposed in this work.

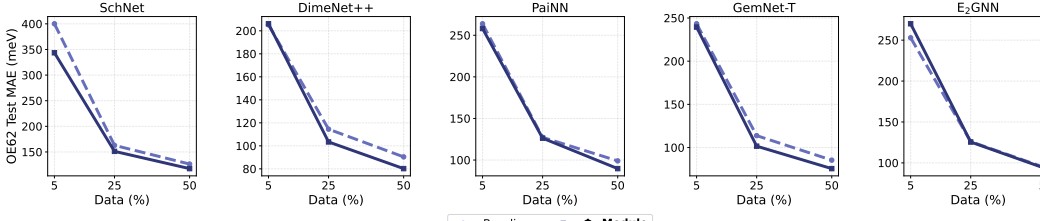

Figure 6: Performance of baseline models and models with Φ-Module in data-scarce setups. Φ-Module outperforms baseline in almost any case for all of the tested setups (5%, 25% and 50% of the initial training data). See Section 4.

**Data Scarcity Configurations.** In this experiment, we demonstrate that Φ-Module achieves performance gains over baselines even in data-scarce cases. We train OE62 baselines and their versions with Φ-Module on *5%, 25%, 50%* of initial training data. Results show that usage of Φ-Module leads to improved performance on nearly every setup and model highlighting stability under various data configurations. Refer to Figure 6 for more details.

## 5 RELATED WORK

**Electrostatic Constraints for Neural Network Potentials.** There are a number of attempts to utilize electrostatic interactions with neural network potentials.

Figure 7: Memory consumption of baseline SchNet, Φ-SchNet, Ewald message passing, and Neural P3M on a carbyne chain with gradually increasing size. Out-of-memory (OOM) points are shown as crosses. Ewald and Neural P3M quickly result in OOM, while the Φ-Module demonstrates strong memory efficiency. See Section 4.

Some of them rely on effective partial charges of atomic nuclei (Xie et al., 2020; Niblett et al., 2021) or incorporate precomputed electronegativities as a starting point (Ko et al., 2023). An alternative approach involves multipole expansion to express electrostatic potentials without reliance on fixed partial-charge approximation (Thürlemann et al., 2021). Although the approach brings performance benefits, it requires expensive training data with information at electronic level.

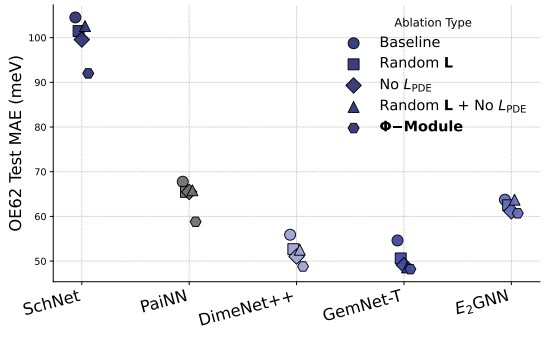

In contrast to mentioned methods, Φ-Module does not require any difficult-to-obtain prior information to deliver valuable improvements for neural networks potentials. Electrostatics are learned in the self-supervised manner using lightweight message passing submodule.

Figure 8: Ablation study on the main design choices for the Φ-Module. We remove structural information from **L** and the optimization of the PDE residual. These "non-physical" variants underperform compared to the baseline and full Φ-Module, highlighting the value of physical priors. See Section 4.

**Ewald Summation.** A separate track of research equips neural network potentials with the electrostatic knowledge via operations related to Ewald summation (Ewald, 1921). For instance, Kosmala et al. (2023) develops Ewald Message Passing, an augmentation to neural message passing with Fourier space interactions and the cutoff in frequency range. Later, Cheng (2024) extrapolated the idea of Particle-Particle-Particle-Mesh (P3M)

(Hockney & Eastwood, 2021) to neural message passing setup, resulting in improved speed compared to regular Ewald Message Passing.

Those approaches focus on the incorporation of the Ewald summation into neural network interatomic potentials, which is an orthogonal line of work. Moreover, originally Ewald summation is constrained to periodic crystals and its usage on non-periodic systems relies on the definition of single large supercell, which serves only as an artificial workaround for such data types. In this paper, we discover a different route of efficient self-supervised learning of electrostatics from Poisson's equation formulation itself.

**General Poisson Learning.** The use of neural networks for solving the Poisson equation began in the mid-1990s, marked by early implementations of multilayer perceptrons to handle the two-dimensional case with Dirichlet boundary conditions (Dissanayake & Phan-Thien, 1994).

In subsequent years, physics-informed neural networks (PINNs) emerged as a powerful approach by embedding the governing differential equations directly into the loss function (Hafezianzade et al., 2023). This methodology has proven especially effective for challenging problems such as the nonlinear Poisson–Boltzmann equation, where traditional numerical methods often struggle with nonlinearity and complex geometries (Mills & Pozdnyakov, 2022).

Recent studies have investigated error correction strategies in neural network-based solvers for differential equations, often using Poisson's equation as a testbed due to its fundamental role as a second-order linear PDE and its broad relevance in theoretical physics (Wright, 2022).

Our work does not aim to solve Poisson's equation explicitly. Instead, we investigate how it can be used to enhance the learning dynamics and performance precision of neural network interatomic potentials.

## 6    Conclusion and Future Work

We introduced the Φ-Module, a universal and physically grounded framework for incorporating electrostatics into neural interatomic potentials. Our method integrates seamlessly with a wide range of deep learning architectures in computational chemistry, offering stable improvements in energy prediction and molecular dynamics tasks. It also demonstrates favorable memory and computational efficiency, with minimal need for hyperparameter tuning. Despite its strengths, the current implementation relies on partial charge approximations and does not yet account for more expressive electrostatic descriptors, such as multipole expansions or polarizability tensors. Additionally, the method is still influenced by a graph connectivity as like any graph-based neural network interatomic potential. Extending the Φ-Module to capture higher-order effects presents a promising direction for advancing self-supervised learning in quantum chemistry.

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

---

**Algorithm 1** Message Passing with $\Phi$-Module

---

**Require:** Mini-batch $\mathcal{B} = \{G_i = (V_i, E_i)\}_{i=1}^{B}$, # message-passing layers $T$, modes $k$
**Ensure:** Total loss $\mathcal{L}$ for back-propagation
1: $L \leftarrow \text{BLOCKDIAG}\big(\{\text{GRAPHLAPLACIAN}(G_i)\}_{i=1}^{B}\big)$
2: $(U, \Lambda) \leftarrow \text{LOBPCG}(L, k)$         ▷ batched eigendecomposition
3: $\{h_v^0\}_{v \in V} \leftarrow \text{EMBEDDING}(\mathcal{B})$
4: $\phi^0 \leftarrow \mathbf{0}; \quad \rho^0 \leftarrow \mathbf{0}$
5: **for** $t = 0$ **to** $T - 1$ **do**
6:      $\{h_v^{t+1}\} \leftarrow \text{MESSAGEPASSING}(\{h_v^t\}, \mathcal{B})$
7:      **if** $t == 0$ **then**
8:          $\alpha_\phi^0, \alpha_\rho^0 \leftarrow \text{ALPHANET}\big(\{h_v^{t+1}\}\big)$
9:          $\phi^1 \leftarrow U \alpha_\phi^0; \quad \rho^1 \leftarrow U \Lambda \alpha_\rho^0$
10:      **else**
11:          $\alpha_\phi^t, \alpha_\rho^t \leftarrow \text{ALPHANET}\big(\{h_v^{t+1}\}\big)$
12:          $\phi^{t+1} \leftarrow \phi^t + U \alpha_\phi^t$
13:          $\rho^{t+1} \leftarrow \rho^t + U \Lambda \alpha_\rho^t$
14:      **end if**
15: **end for**
16: $\mathbf{E}^{\text{model}} \leftarrow \text{READOUTENERGY}\big(\{h_v^T\}, \mathcal{B}\big)$
17: $\mathbf{E}^{\text{ES}} \leftarrow \frac{1}{2} \sum_{v \in V_i}(\phi_v \rho_v)$         ▷ electrostatic energy term
18: $\mathbf{r} \leftarrow L \phi^T - \rho^T$         ▷ PDE residual
19: $\mathcal{L} \leftarrow \underbrace{\ell(\mathbf{E}^{\text{model}} + \mathbf{E}^{\text{ES}}, \mathbf{E}^{\text{target}})}_{\mathcal{L}_{\text{model}}} + \beta \underbrace{\|\mathbf{r}\|_2}_{\mathcal{L}_{\text{PDE}}} + \gamma \underbrace{|\sum_{v \in V_i} \rho_v^T|}_{\mathcal{L}_{\text{net}}}$
20: **return** $\mathcal{L}$

---

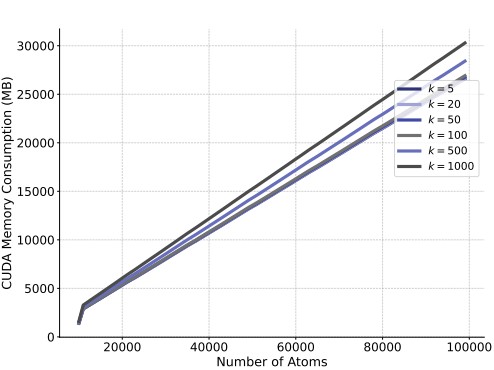

(a) Memory scaling of $\Phi$-Module with varying $k$.

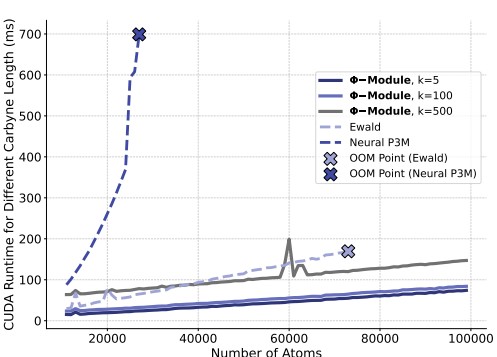

(b) GPU runtime comparison of $\Phi$-Module with Ewald and Neural P3M.

Figure 9: Additional memory and runtime scaling experiments.

## A  CODE AVAILABILITY

We provide the source code to reproduce the experiments in the supplementary material to the submission as a file archive. The code will be released to public upon acceptance.

## B  PSEUDOCODE FOR $\Phi$-MODULE

Complete and detailed pseudocode for $\Phi$-Module can be examined in Algorithm 1.

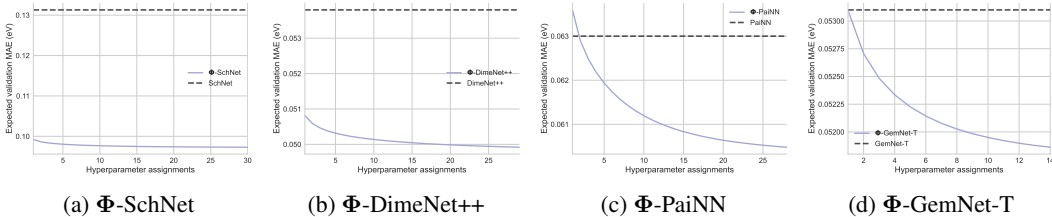

| (a) $\Phi$-SchNet | (b) $\Phi$-DimeNet++ | (c) $\Phi$-PaiNN | (d) $\Phi$-GemNet-T |

Figure 10: Energy–variance plots for $\Phi$-variants.

Table 2: $\Phi$-Module hyperparameters for the reported models on OE62.

| Model | Hyperparameters | | |
|---|---|---|---|
| | $k$ | $\beta$ | $\gamma$ |
| $\Phi$-SchNet | 9 | $10^{-4}$ | $10^{-4}$ |
| $\Phi$-DimeNet++ | 9 | $10^{-2}$ | $10^{-4}$ |
| $\Phi$-PaiNN | 10 | $10^{-3}$ | $10^{-1}$ |
| $\Phi$-GemNet-T | 3 | $5*10^{-1}$ | $10^{-3}$ |
| $\Phi$-E$_2$GNN | 5 | $10^{-1}$ | 0 |

## C  MEMORY AND RUNTIME SCALING WITH INCREASING $k$.

In this experiment, we follow the design of the task involving the linear carbyne chain in Section 4, but test the memory and runtime trends of $\Phi$-Module with respect to the increasing number of estimated eigenvalues. In Figure 9a, you can see that memory consumption increases at a very slow rate which allows efficient processing of large systems. Moreover, $\Phi$-Module scales favorably in terms of GPU runtime for large systems (starting from $10^4$ atoms) in comparison to Ewald and Neural P3M based on Figure 9b.

## D  HYPERPARAMETERS

**Hyperparameter Search.**  We run a hyperparameter search with random uniform sampling for the $\Phi$-Module with the following configuration for each model in this study: $k$: $\{3, 5, 7, 9, 10, 15\}$, $\beta$: $\{10^{-4}, 10^{-3}, 10^{-2}, 10^{-1}, 5*10^{-1}\}$, $\gamma$: $\{10^{-4}, 10^{-3}, 10^{-2}, 10^{-1}, 5*10^{-1}\}$. All experiments were conducted on one NVIDIA 80G H100 GPU.

**OE62.**  We follow the same hyperparameters for the baselines as in Kosmala et al. (2023) in most cases. The main difference are the usage of Adam (Kingma & Ba, 2014) optimizer and cosine learning rate schedule (Loshchilov & Hutter, 2016) without warm restarts as well as gradient clipping of $10^3$. For hyperparameters related to $\Phi$-Module, refer to Table 2. For E$_2$GNN we use 256 hidden channels, 6 layers, 128 Gaussian RBFs, cutoff of 6.0 Å with a maximum 50 neighbors. Batch size is set to 64 and the training of E$_2$GNN was performed for 400 epochs with the same optimizer and scheduler as for other models without gradient clipping.

**MD22.**  For the baseline ViSNet, we employ the same hyperparameters as in (Wang et al., 2022). Optimizer and schduler choice is the same as for OE62. The hyperparameters for $\Phi$-ViSNet are the following: $k$ - 9, $\beta$ - $10^{-3}$, $\gamma$ - $10^{-4}$.

## E  OE62 RESULTS

In Table 3 the exact numerical comparison for the Section 4 OE62 experiment is shown.

Table 3: Energy MAEs and computation time of baselines and their alternatives with Ewald module and Φ-Module on OE62. Lowest errors and fastest runtimes compared to baselines are highlighted in bold. See Section 4.

| Model | Version | OE62-val MAE, meV ↓ | OE62-val Rel, % ↓ | OE62-test MAE, meV ↓ | OE62-test Rel, % ↓ | Mean Epoch Time Runtime, s ↓ | Mean Epoch Time Rel, % ↓ |
|---|---|---|---|---|---|---|---|
| SchNet | Baseline | 133.5 | - | 131.3 | - | 16.91 | - |
| | Ewald | **79.2** | 40.7 | **81.1** | 38.2 | 68.4 | 304.5 |
| | Φ-SchNet | 92.2 | 30.9 | 92.0 | 29.9 | **18.65** | 10.3 |
| DimeNet++ | Baseline | 51.2 | - | 53.8 | - | 90.66 | - |
| | Ewald | **46.5** | 9.2 | **48.1** | 10.6 | 212.1 | 134.0 |
| | Φ-DimeNet++ | 47.1 | 8.0 | 48.8 | 9.3 | **94.36** | 4.1 |
| PaiNN | Baseline | 61.4 | - | 63.0 | - | 56.70 | - |
| | Ewald | 57.9 | 5.7 | 59.7 | 5.2 | 193.2 | 240.9 |
| | Φ-PaiNN | **57.7** | 6.0 | **58.8** | 6.7 | **62.24** | 9.8 |
| GemNet-T | Baseline | 51.2 | - | 53.1 | - | 179.01 | - |
| | Ewald | **46.5** | 9.2 | **47.5** | 10.5 | 501.0 | 179.9 |
| | Φ-GemNet-T | 47.3 | 7.6 | 48.2 | 9.2 | **187.40** | 4.7 |
| E$_2$GNN | Baseline | 60.9 | - | 61.6 | - | 130.8 | - |
| | Ewald | 60.3 | 1.0 | 61.0 | 1.0 | 185.1 | 41.5 |
| | Φ-E$_2$GNN | **59.2** | 2.8 | **60.7** | 1.5 | **162.0** | 23.9 |

The results show that the Φ-Module provides consistent performance improvements across all baselines, with gains of at least 5% in most cases and around 3% for E$_2$GNN. In addition, it outperforms the Ewald block in 2 out of 5 settings while requiring noticeably less computational overhead.

# F   ARCHITECTURAL DETAILS OF $\alpha$-NET

**Permutational Invariance.**   The convolutions in the $\alpha$-Net are applied over the node dimension. The permutational invariance is lost only given the kernel size is not 1. Convolutions with 1x1 filter can also serve as a competitive option. Below (see Table 4) are results comparing SchNet and DimeNet++ with regular $\alpha$-Net and the one with 1x1 convolutions preserving invariance. Both options show similar performance.

**Separate Eigenbasis Coefficients $\alpha_\phi$ and $\alpha_\rho$.**   In this paragraph, we discuss the idea of learning separated Laplacian eigenbasis coefficients for potential and charges. Theorem F.1 describes the symmetric nature of residual gradient w.r.t $\alpha_\phi$ and $\alpha_\rho$, given the parametrization of $\rho = U\Lambda\alpha_\rho$ aligned with the eigenbasis of $L\phi = U\Lambda U^\top U\alpha_\phi = U\Lambda\alpha_\phi$. We expect this parametrization to benefit training dynamics as such symmetries guarantee equal update rate for both $\alpha_\phi$ and $\alpha_\rho$. On the other hand, plain $\rho = U\alpha_\rho$ results in the dependence on $\lambda_i$ making an optimization process dominated by specific modes and neglecting others.

**Proposition F.1** (Symmetric vs. asymmetric gradients for the Poisson residual). *Preserving the notations from Theorem 3.1, let the potential be $\phi = U\alpha_\phi$ and define the Poisson residual loss*

$$\mathcal{L}(\alpha_\phi, \alpha_\rho) = \beta \, \|L\phi - \rho\|_2^2, \qquad \beta > 0.$$

*Then:*

*Case (A): ($\rho = U\Lambda\alpha_\rho$). Writing $r = L\phi - \rho = U\Lambda(\alpha_\phi - \alpha_\rho)$, the gradients are*

$$\nabla_{\alpha_\phi}\mathcal{L} = 2\beta\,\Lambda^2(\alpha_\phi - \alpha_\rho), \qquad \nabla_{\alpha_\rho}\mathcal{L} = -2\beta\,\Lambda^2(\alpha_\phi - \alpha_\rho).$$

*Per mode $i$:*

$$\frac{\partial\mathcal{L}}{\partial(\alpha_\phi)_i} = 2\beta\,\lambda_i^2((\alpha_\phi)_i - (\alpha_\rho)_i), \quad \frac{\partial\mathcal{L}}{\partial(\alpha_\rho)_i} = -2\beta\,\lambda_i^2((\alpha_\phi)_i - (\alpha_\rho)_i).$$

*Hence the updates are equal in magnitude and opposite in sign, with identical per-mode scaling $\lambda_i^2$ resulting in mode-wise symmetry.*

***Case (B):*** *($\rho = U\alpha_\rho$). Writing $r = L\phi - \rho = U(\Lambda\alpha_\phi - \alpha_\rho)$, the gradients are*

$$\nabla_{\alpha_\phi}\mathcal{L} = 2\beta\,\Lambda(\Lambda\alpha_\phi - \alpha_\rho), \qquad \nabla_{\alpha_\rho}\mathcal{L} = -2\beta\,(\Lambda\alpha_\phi - \alpha_\rho).$$

*Per mode $i$:*

$$\frac{\partial\mathcal{L}}{\partial(\alpha_\phi)_i} = 2\beta\,\lambda_i(\lambda_i(\alpha_\phi)_i - (\alpha_\rho)_i)\,, \quad \frac{\partial\mathcal{L}}{\partial(\alpha_\rho)_i} = -2\beta(\lambda_i(\alpha_\phi)_i - (\alpha_\rho)_i)\,.$$

*Thus the two updates differ by a factor $\lambda_i$ resulting in mode-wise asymmetry.*

*Proof.* Let $r = L\phi - \rho$. Then, $\nabla_\phi\|r\|_2^2 = 2L^\top r = 2Lr$ and $\nabla_\rho\|r\|_2^2 = -2r$. Mapping to $\phi = U\alpha_\phi$ and applying the chain rule we acquire $\nabla_{\alpha_\phi}\|r\|_2^2 = U^\top(2Lr)$.

*Case (A).* If $\rho = U\Lambda\alpha_\rho$, then $r = U\Lambda(\alpha_\phi - \alpha_\rho)$ and

$$\nabla_{\alpha_\phi}\mathcal{L}_{\text{PDE}} = \beta\,(U\Lambda)^\top(2r) = 2\beta\,\Lambda U^\top U\Lambda(\alpha_\phi - \alpha_\rho) = 2\beta\,\Lambda^2(\alpha_\phi - \alpha_\rho),$$

$$\nabla_{\alpha_\rho}\mathcal{L}_{\text{PDE}} = \beta\,(-U\Lambda)^\top(2r) = -2\beta\,\Lambda^2(\alpha_\phi - \alpha_\rho).$$

*Case (B).* If $\rho = U\alpha_\rho$, then $r = U(\Lambda\alpha_\phi - \alpha_\rho)$ and

$$\nabla_{\alpha_\phi}\mathcal{L}_{\text{PDE}} = \beta\,(U\Lambda)^\top(2r) = 2\beta\,\Lambda(\Lambda\alpha_\phi - \alpha_\rho), \qquad \nabla_{\alpha_\rho}\mathcal{L}_{\text{PDE}} = \beta\,(-U)^\top(2r) = -2\beta\,(\Lambda\alpha_\phi - \alpha_\rho).$$

$\square$

**Nature of the Laplacian.** For a molecular graph $G = (V, E)$ with node set $V$ and positive symmetric edge weights $w_{ij} > 0$, we employ the *symmetric normalized Laplacian*

$$L\ =\ I - D^{-\frac{1}{2}}WD^{-\frac{1}{2}},$$

where $W \in \mathbb{R}^{|V|\times|V|}$ is the weighted adjacency matrix with entries $W_{ij} = w_{ij}$ if $(i, j) \in E$ and 0 otherwise, and $D = \text{diag}(d_1, \ldots, d_{|V|})$ is the diagonal degree matrix with $d_i = \sum_j W_{ij}$. By construction $L$ is real, symmetric, and positive semidefinite. We use the edge weights $w_{ij} = d_{ij}$ given by the interatomic distances between atoms $i$ and $j$, which preserves symmetry and ensures that $L$ encodes geometric information about molecular conformations.

# G ADDITIONAL BACKGROUND.

**Microcanonical Ensemble.** In classical molecular dynamics, the microcaninical ensemble (NVE) models an isolated system with constant particle number ($N$), volume ($V$), and total energy ($E$). The dynamics follow Newton's equations of motion:

$$m_i\ddot{\mathbf{r}}_i(t)\ =\ -\nabla_{\mathbf{r}_i}U(\mathbf{r}_1, \ldots, \mathbf{r}_N), \tag{5}$$

where $\mathbf{r}_i(t)$ denotes the position of particle $i$, $m_i$ its mass, and $U$ the potential energy function. In the absence of thermostats or external driving, this guarantees conservation of total energy.

To integrate trajectories numerically, Verlet-type schemes are widely adopted due to their symplecticity and time reversibility. The basic Verlet update is given by

$$\mathbf{r}_i(t + \Delta t)\ =\ 2\mathbf{r}_i(t) - \mathbf{r}_i(t - \Delta t) + \frac{\Delta t^2}{m_i}\mathbf{F}_i(t), \tag{6}$$

with forces $\mathbf{F}_i(t) = -\nabla_{\mathbf{r}_i}U$. A more practical variant is the velocity Verlet integrator:

$$\mathbf{r}_i(t + \Delta t) = \mathbf{r}_i(t) + \Delta t\,\mathbf{v}_i(t) + \tfrac{1}{2}\Delta t^2\,\mathbf{a}_i(t), \tag{7}$$

$$\mathbf{v}_i(t + \Delta t) = \mathbf{v}_i(t) + \tfrac{1}{2}\Delta t\big(\mathbf{a}_i(t) + \mathbf{a}_i(t + \Delta t)\big), \tag{8}$$

where $\mathbf{a}_i(t) = \mathbf{F}_i(t)/m_i$. These integrators achieve $\mathcal{O}(\Delta t^2)$ accuracy while requiring only a single force evaluation per timestep. Crucially, their symplectic structure ensures stable long-time energy behavior, making NVE with Verlet the de facto baseline.

Table 4: Comparison of $\alpha$-net variants for SchNet and DimeNet++.

| MODEL | DEFAULT $\alpha$-NET | $1\times1$ CONV $\alpha$-NET |
|-------|------------------|--------------------------|
| SCHNET | **92.0** | 93.8 |
| DIMENET++ | **48.8** | 49.5 |

**Expected Validation Performance.** Expected Validation Performance (EVP) Dodge et al. (2019) curve represents how on average performance changes as the number of hyperparameter assignments increases during the random search. The X-axis represents the number of hyperparameter trials. The Y-axis represents the expected best performance for a given number of hyperparameter trials.

The expected best performance is computed as

$$\mathbb{E}[V_n^*|n] = \sum_v v \cdot (P(V_i \leq v)^n - P(V_i < v)^n),$$

where $V_n^* = \max_{i \in \{1,\ldots,n\}} V_i$ is the maximum for model performance evaluations $V_i$ given a series of $n$ i.i.d. hyperparameter configurations, which are acquired empirically from the random hyperparameter search process and $P(V_n^*|n)$ is the probability mass function for the max-random variable.

The EVP curves for $\Phi$-SchNet, $\Phi$-DimeNet++, $\Phi$-PaiNN and $\Phi$-GemNet-T can be seen in **??**. $\Phi$-Module demonstrates hyperparameter stability for all of the baseline models.

you removed all text completely. listen again, make it as close as possible to the initial version below

here are proofs for theorems, check briefly if they are correct

## H   PROOFS

In this part we restate Theorem 3.1 and Theorem 3.2 from the main body and proof them accordingly. Note that we prove theorems for the surrogate L2 objective for the tractability, and it is interchangeable.

**Theorem H.1** (Exact inner minimizer over $\rho$). *Define $a = \mathrm{E} - \mathrm{E}_{model}$. Fix $\phi \in \mathrm{span}(U_k)$. The unique minimizer of $\rho \mapsto \mathcal{L}(\phi, \rho)$ over $\mathrm{span}(U_k)$ is*

$$\rho^\star(\phi) = L\phi - t^\star(\phi)\,\phi, \qquad t^\star(\phi) = \frac{a + \frac{1}{2}\,\phi^\top L\phi}{2\beta + \frac{1}{2}\,\|\phi\|^2}.$$

*Proof.* Let $e(\phi, \rho) := a + \frac{1}{2}\,\phi^\top \rho$. Using $\nabla_\rho \|L\phi - \rho\|^2 = -2(L\phi - \rho)$ and $\nabla_\rho\, e(\phi, \rho)^2 = 2e(\phi, \rho) \cdot \frac{1}{2}\phi = e(\phi, \rho)\phi$, the first-order condition is

$$\nabla_\rho \mathcal{L}(\phi, \rho) = 2\beta(\rho - L\phi) + e(\phi, \rho)\,\phi = 0.$$

Hence $\rho - L\phi$ is colinear with $\phi$, then $\rho = L\phi - t\phi$ for some $t \in \mathbb{R}$. Substituting back gives

$$-2\beta t\phi + \left(a + \frac{1}{2}(\phi^\top L\phi - t\|\phi\|^2)\right)\phi = 0,$$

and we obtain $-2\beta t + a + \frac{1}{2}\phi^\top L\phi - \frac{1}{2}t\|\phi\|^2 = 0$, i.e.

$$t^\star(\phi) = \frac{a + \frac{1}{2}\,\phi^\top L\phi}{2\beta + \frac{1}{2}\,\|\phi\|^2}.$$

Uniqueness follows because the Hessian w.r.t. $\rho$ is $2\beta I + \frac{1}{2}\,\phi\phi^\top \succ 0$ for $\beta > 0$. $\qquad\square$

**Theorem H.2** (Monotone objective decrease in optimization towards $\rho^\star$). *Define $A(\phi) := a + \frac{1}{2}\phi^\top L\phi$. Then substituting $\rho^\star(\phi)$ from Theorem 3.1 yields*

$$\widetilde{\mathcal{L}}(\phi) := \mathcal{L}(\phi, \rho^\star(\phi)) = A(\phi)^2\,\frac{4\beta}{4\beta + \|\phi\|^2} \leq A(\phi)^2,$$

*with equality if and only if $A(\phi) = 0$ or $\phi = 0$.*

*Proof.* Along the affine line $\rho(t) = L\phi - t\phi$ we have

$$\mathcal{L}(\phi, \rho(t)) = \beta \|t\phi\|^2 + \left(A(\phi) - \tfrac{1}{2}t\|\phi\|^2\right)^2 = \underbrace{\beta\|\phi\|^2}_{p} t^2 + \left(A(\phi) - \underbrace{\tfrac{1}{2}\|\phi\|^2}_{q} t\right)^2.$$

This is a strictly convex quadratic in $t$ (for $\|\phi\|^2 > 0$) with minimizer $t^\star = \frac{A(\phi)q}{p+q^2}$ and minimum value

$$\mathcal{L}\big(\phi, \rho^\star(\phi)\big) = A(\phi)^2 \frac{p}{p+q^2} = A(\phi)^2 \frac{\beta\|\phi\|^2}{\beta\|\phi\|^2 + \tfrac{1}{4}\|\phi\|^4} = A(\phi)^2 \frac{4\beta}{4\beta + \|\phi\|^2}.$$

Since $\frac{4\beta}{4\beta+\|\phi\|^2} \in (0, 1]$, the inequality follows; equality holds exactly when $A(\phi) = 0$ or $s(\phi) = 0$. $\qquad\square$

