# OpenReview forum: "Electrostatics from Laplacian Eigenbasis for Neural Network Interatomic Potentials"
_ICLR.cc/2026/Conference — Submitted to ICLR 2026_

### Official Review · Reviewer_QJiW · 2025-10-15

**Soundness:** 3
**Presentation:** 3
**Contribution:** 3
**Rating:** 2
**Confidence:** 5

**Summary:**

This paper introduces Φ-Module, a plug-and-play Laplacian-eigenbasis module that embeds the Poisson equation as a self-supervised physical constraint within graph neural network (GNN) interatomic potentials.

**Strengths:**

The proposed module can predict atomic potential ϕ and charge ρ from learned eigenbasis coefficients and adds an electrostatic energy term to improve total-energy and force predictions.

**Weaknesses:**

The paper repeatedly claims that Φ-Module captures non-local electrostatic interactions, yet no benchmark explicitly demonstrates this. Improvements in energy and force MAE alone are insufficient proof of non-local interaction modeling.

For MD22, the authors selected two of the smallest molecules, while the claim centers on modeling non-local interactions. Larger systems such as Ac-Ala15-NHMe or DHA are more appropriate to evaluate the alleged long-range capability. Without them, the argument remains speculative.

On the OE62 experiments, the baselines (SchNet, DimeNet++, PaiNN, etc.) are 2019–2021 architectures. Recent high-order equivariant models—MACE, eSCN, NequIP, Equiformer-V2, ViSNet—represent the current state of the field. Because the Φ-Module is advertised as a general plug-in, results on these modern architectures are essential for a credible evaluation.

On the MD22 experiments, the reported performance differences from the original ViSNet paper are unusually large for the same datasets. The authors should clarify training settings (data splits, learning rates, cutoff, etc.) and reproduce the original ViSNet baseline under identical conditions.

**Questions:**

See Section Weakness.

---

> ### Author Response · Authors · 2025-11-22
> **Rebuttal by Authors**
>
> We are grateful for the reviewer’s attention to details. We address mentioned points below.
>
> **Q: The paper repeatedly claims that Φ-Module captures non-local electrostatic interactions, yet no benchmark explicitly demonstrates this. Improvements in energy and force MAE alone are insufficient proof of non-local interaction modeling**
>
> A: We thank the reviewer for the important clarification. We are currently in the process of training models on million-size large OMol25 dataset which will hopefully show how Φ-Module helps treating molecules with diverse long-range interaction. We will post results soon. We thank the reviewer for the patience!
>
> **Q: On the OE62 experiments, the baselines (SchNet, DimeNet++, PaiNN, etc.) are 2019–2021 architectures. Recent high-order equivariant models—MACE, eSCN, NequIP, Equiformer-V2, ViSNet—represent the current state of the field. Because the Φ-Module is advertised as a general plug-in, results on these modern architectures are essential for a credible evaluation.**
>
> A: We appreciate reviewer’s suggestion. We computed results for E2GNN (Yang, Ziduo, et al. "Efficient equivariant model for machine learning interatomic potentials." *npj Computational Materials* 11.1 (2025): 49.) in Table 1, which demonstrate Φ-Module on a recent architecture. We would like to additionally insert other models
>
> **Q: On the MD22 experiments, the reported performance differences from the original ViSNet paper are unusually large for the same datasets. The authors should clarify training settings (data splits, learning rates, cutoff, etc.) and reproduce the original ViSNet baseline under identical conditions.**
>
> A: We appreciate the opportunity to discuss experiment design. We reproduced ViSNet results using original code and training splits as can be noted in our code and indeed we observed several inconsistencies. Nevertheless, we still report favorable performance of Φ-Module under this setting. We would like to report additional updated numbers for several MD22 molecules if the time allows.

---

### Official Review · Reviewer_oDat · 2025-10-31

**Soundness:** 2
**Presentation:** 2
**Contribution:** 3
**Rating:** 4
**Confidence:** 4

**Summary:**

This work introduces $\Phi$-Module, an extension to atom-based machine learning interatomic potentials that intends to resolve long-range energy contributions. $\Phi$-Module is derived from the discretized Poisson equation for electrostatics and uses the graph Laplacian for efficient long-range propagation. In their experimental evaluation, the authors find the $\Phi$-Module to be an effficient addition to MLIAP, reducing errors at low computational overheads.

**Strengths:**

* The proposed $\Phi$-module presents a novel addition to the field of MLFFs.
* The experiments suggest a very valuable extensions with favorable runtime-accuracy tradeoff.
* $\Phi$-module appears quite modular and presents an easy integration.

**Weaknesses:**

1. The authors claim that their method is self-supervised and does not need external labeled data (l.50-51 + abstract). However, the losses introduced in l.174 and l.179 do not ensure that predictions improve, the method additionally **needs** the standard supervised learning loss.
2. The use of 1D convolutions over the nuclei breaks the permutation invariance. I find the paper lacks to discuss this disadvantage. One could in general neglect permutation invariance in MLFFs, does that yield similar improvements?
3. The authors leave a lot of questions open or are not specific in several key areas, see questions.


Minor:
* l.26-33: DFT and MLFFs are quite different things and the section suggest they accomplish the same.
* l.34 which alternatives?
* Table 1 is never referenced.
* l.432 formatting

**Questions:**

1. What is used as graph connectivity for the graph laplacian? A radial cutoff or molecular bonds? In either case, doesn't bond-breaking or going out of the cutoff radius, introduces jumps in the energy surface?
2. Why is the Laplacian weighted by the pairwise distance, intuitively, I'd expect it to be weighted by the inverse of the distance?
3. l.173 is the same alpha-Net used for phi and rho?
4. Couldn't one enforce the PDE and net zero loss analytically? How does that compare to training?
5. What is the x-axis in Figure 4? (Also the figure is never referenced)
6. Table 1: What about other MLFF+Phi module combinations?
7. Are the hyperparameter optimization from l.684 used for all experiments? This seems quite excessive given that no hyperparameter optimization is done for any of the baseline models.
8. What units are Table 4 and what target, which dataset, etc?

---

> ### Author Response · Authors · 2025-11-22
> **Rebuttal by Authors**
>
> We are grateful for the reviewer’s attention to details. We address mentioned points below.
>
> **Q:  The authors claim that their method is self-supervised and does not need external labeled data (l.50-51 + abstract). However, the losses introduced in l.174 and l.179 do not ensure that predictions improve, the method additionally needs the standard supervised learning loss.**
>
> A: We thank the reviewer for the opportunity to clarify this point. The term “self-supervised” in the paper refers specifically to the learning of the electrostatic quantities (Equation 3 and 4). Φ-Module does not require any external labels for partial charges. Those quantities are derived purely from the Poisson constraint (Equation 2), the Laplacian eigenbasis and the model’s hidden representations. At the same time, Φ-Module is integrated into a standard neural interatomic potential, which is trained using supervised losses on energies and forces like all NNIPs. This is the loss referenced in lines 174 and 179.
>
> **Q: The use of 1D convolutions over the nuclei breaks the permutation invariance. I find the paper lacks to discuss this disadvantage. One could in general neglect permutation invariance in MLFFs, does that yield similar improvements?**
>
> A: α-Net applies convolutions over node dimension. However, the permutational invariance is lost only given the kernel size is not 1. Convolutions with 1x1 filter (simply a linear layer) also can serve as a competitive option. Below are results comparing SchNet and DimeNet++ with regular α-Net and the one with 1x1 convolutions preserving invariance. Both options show similar performance. Results are presented in Appendix F.
>
> **Q: What is used as graph connectivity for the graph laplacian? A radial cutoff or molecular bonds? In either case, doesn't bond-breaking or going out of the cutoff radius, introduces jumps in the energy surface?**
>
> A: For the graph Laplacian we use a radius graph based on atomic positions, with the same cutoff as the underlying NNIP. In practice, we choose a cutoff value (5-6 Å) such that chemically relevant interactions are far inside the cutoff, so the corresponding change in energy is small. Moreover, our MD stability experiments (NVE/NVT on MD22 systems, Figure 5) show that Φ-ViSNet has stable trajectories and in some cases improved structural stability compared to ViSNet, indicating that cutoff effects are not problematic in the regimes we study.
>
> **Q: Why is the Laplacian weighted by the pairwise distance, intuitively, I'd expect it to be weighted by the inverse of the distance?**
>
> A: We welcome the chance to clarify this implementation detail of the Φ-Module. A graph Laplacian is designed to approximate the negative Laplacian operator, so its weights must capture local geometric structure. Additionally, the Φ-Module does not encode Coulomb interactions through adjacency weights. Instead, Coulomb behavior emerges from solving the Poisson equation under the constraints defined in Equation 2. Empirically, alternative schemes such as inverse-distance scaling were tested but produced substantially weaker performance (explore table below).
>
> |  | OE62-Val | OE62-Test |
> | --- | --- | --- |
> | Φ-SchNet | 92.2 | 92.0 |
> | Φ-SchNet with 1/d | 112.3 | 113.5 |
>
> **Q: l.173 is the same alpha-Net used for phi and rho?**
>
> A: No, for phi and rho, α-Nets share the backbone, but have different heads to avoid degeneracy in self-supervised learning.
>
> **Q: Couldn't one enforce the PDE and net zero loss analytically? How does that compare to training?**
>
> A: We thank the reviewer for this question. Because ϕ and ρ are formed by independent contributions from multiple α-vectors, there is no closed-form solution for α that makes the discrete Poisson equation hold identically. As a result, the residual $L\phi  - \rho$ cannot be forced to zero analytically, and the PDE loss is necessary to guide the optimization. Ablations in Figure 8 show that removing the PDE residual leads to worse accuracy, confirming that analytic PDE enforcement is neither possible nor desirable.
>
> **Q: What is the x-axis in Figure 4? (Also the figure is never referenced)**
>
> A: X-axis in Figure 4 refers to number of hyperparameter configurations one needs to test on average to get the excepted validation MAE. That is Expected Validation Performance from  Dodge et al., 2019.

---

> > ### Author Response · Authors · 2025-11-22
> > **Rebuttal by Authors**
> >
> > **Q: Table 1: What about other MLFF+Phi module combinations?**
> >
> > A:  We appreciate the reviewer’s question. In the MD22 experiments, we use ViSNet as the primary baseline for Φ-Module for two reasons. Firstly, ViSNet was tested on MD22 explicitly and currently is one of the most accurate and efficient equivariant NNIPs for molecular dynamics due to its low-cost of geometrical message passing. Secondly, MD22 requires extensive GPU resources, making a evaluation of multiple models very expensive.
> >
> > **Q: Are the hyperparameter optimization from l.684 used for all experiments? This seems quite excessive given that no hyperparameter optimization is done for any of the baseline models.**
> >
> > A: We thank the reviewer for this question. Optimizing only the Φ-specific hyperparameters while keeping all baseline settings fixed ensures that our comparisons isolate the contribution of Φ-Module itself, without re-tuning the underlying architectures. Additionally, we demonstrate how almost any combination of hyperparameters results in improvements in energy prediction (Figure 4).
> >
> > **Q: What units are Table 4 and what target, which dataset, etc?**
> >
> > A: Units for Table 4 are the same as for Table 1. It is MAE in meV on OE62 test for energy prediction.
> >
> > Additionally we would like to thank the reviewer for detailed formatting analysis. We would address all of those corrections in the following version of the paper.

---

### Official Review · Reviewer_H2RH · 2025-11-01

**Soundness:** 3
**Presentation:** 3
**Contribution:** 3
**Rating:** 6
**Confidence:** 4

**Summary:**

This paper presents a module for learning the electrostatic interactions for interatomic potentials based on sparse graph neural networks and Poisson equation.

**Strengths:**

- The proposed method is simple yet performative.
- The paper is very well presented with the advantage of the designed model clearly highlighted.
- The tradeoff between efficiency and performance is discussed in detail.
- The problem addressed in this paper is of importance to the molecular modeling community.

**Weaknesses:**

- There seems to be limited novelty in the proposed graph representation module.

**Questions:**

- Could you kindly elaborate more on the choice of using spectral graph neural networks, rather than the spatial ones, which are more popular?

---

> ### Author Response · Authors · 2025-11-22
> **Rebuttal by Authors**
>
> We thank the reviewer for important feedback. We address it below.
>
> **Q: There seems to be limited novelty in the proposed graph representation module.**
>
> A: We thank the reviewer for critical assessment of Φ-Module. Φ-Module brings novelty due to its self-supervised nature and electrostatic derivation purely on the learning basis. It additionally demonstrates favorable time and memory efficiency compared to related methods.
>
> **Q: Could you kindly elaborate more on the choice of using spectral graph neural networks, rather than the spatial ones, which are more popular?**
>
> A: We thank the reviewe for the insighful question. We are not using a spectral graph neural network. Φ-Module is fully integrated into standard spatial message-passing architectures (SchNet, DimeNet++, PaiNN, GemNet-T, E2GNN, ViSNet). The spectral component is a lightweight regularization module that enforces a Poisson constraint via a truncated Laplacian eigenbasis.

---

### Official Review · Reviewer_gZro · 2025-11-02

**Soundness:** 3
**Presentation:** 3
**Contribution:** 2
**Rating:** 6
**Confidence:** 3

**Summary:**

The paper introduces a Phi module, a module that can be added to any MLIP architecture to introduce a long-range electro statics inductive bias. They imitate the computational structure of a poission equation to learn a sort of latent charges that can be trained self-consistently.

**Strengths:**

-The method is well motivated and makes intuitive sense
-The experimental evaluation is thorough
-The improvements are very consistent

**Weaknesses:**

-The accuracy improvements are rather small
-The spectral decomposition introduces an N^2 scaling operation, which could become problematic for larger-scale simulations. The paper only benchmarks memory, but not runtime with system size; this should be benchmarked and could change my opinion
-There have been works before that incorporate explicit charge equilibrium/coulomb interactions, in particular https://www.nature.com/articles/s41467-020-20427-2 . A comparison would be appropriate
- The phi module only targets electrostatics and not other long range effects, which may make fully learned approaches preferable

**Questions:**

- Do you use a dense connectivity graph L? And why is it weighted by d_ij, doenst this imply that far away atoms interact stronger than closer ones?
- Why did you use VisNet in favour of a newer architecture?

Remark:
"In quantum chemistry, the task of correct prediction of atomic energies is paramount, but stands
a great challenge…” Atomic energies are not a well defined concept, did you mean molecular energies?
“Some of those require prior data in the form of partial charges or
dipole moments, which is costly to retrieve using DFT”, If the DFT calculation is already converged to get the molecular energies, it is trivial to get dipoles or partial charges at negligible costs
“...and gives the opportunity to process large macromolecules with the Φ-Module. This decision also keeps us away from the ambiguity of invariance and sorting of eigenvalues and eigenvectors during their computations - we strictly get k-selected eigenvalues and their corresponding eigenvectors without the need to sort them anyhow.” This is not convincing, if the Laplacina has a degenerate eigenspace the order will be arbitrary and specifics will be subject to numerical noise, a conjugated gradient solver doesnt change this

---

> ### Author Response · Authors · 2025-11-22
> **Rebuttal by Authors**
>
> We sincerely thank the reviewer for noting crucial points regarding the method. We address them below.
>
> **Q: The accuracy improvements are rather small**
>
> A: We fully agree with the reviewer general view of accuracy improvements. Thus, we are computing results for diverse and large-scale (4 million samples) OMol25 for SchNet (small model) and GemNet-O (large model). We thank the reviewer for patience!
>
> **Q: The spectral decomposition introduces an N^2 scaling operation, which could become problematic for larger-scale simulations. The paper only benchmarks memory, but not runtime with system size**
>
> A: We thank the reviewer for questions regarding computational complexity. We provide computational comparison in terms of both, speed and memory, in Appendix C. Φ-Module outperforms other long-range method like Ewald summation or Neural-P3M in efficiency.
>
> **Q: There have been works before that incorporate explicit charge equilibrium/coulomb interactions (4D-HDNNP)**
>
> A: We thank the reviewer for addressing related work. Fourth-generation high-dimensional neural network potential (4G-HDNNP) employ supervised Hirshfeld charges directly from DFT. Φ-Module is a plug-in module for any GNN and is completely self-supervised introducing minimal computational overhead compared to alternatives. We discuss conceptual similarities and differences, but a quantative comparison to 4G-HDNNP would be unfair and out of scope for the mentioned reasons.
>
> **Q: Do you use a dense connectivity graph L? And why is it weighted by d_ij, doenst this imply that far away atoms interact stronger than closer ones?**
>
> A: We appreciate the opportunity to answer this subtle detail about the implementation of Φ-Module. A graph Laplacian is supposed to approximate the negative Laplacian operator and its weights must reflect local geometry. That is the reason on the usage of distance as weights. As an additional clarification, Φ-Module does not model Coulomb physics via adjacency weights, it emerges from Poisson equation itself by satisfying constraints of Equation 2. As an empirical results, we experimented with inverse distance scaling and it produces far weaker results
>
> |  | OE62-Val | OE62-Test |
> | --- | --- | --- |
> | Φ-SchNet | 92.2 | 92.0 |
> | Φ-SchNet with 1/d | 112.3 | 113.5 |
>
> **Q: Why did you use VisNet in favour of a newer architecture?**
>
> A: VisNet had well-established results on molecular dynamics datasets and was employed in previous studies for MD22. It also had clean implementation. Those were the main factors for this choice.
>
> **Q: “Atomic energies are not a well defined concept, did you mean molecular energies?”.**
>
> A: Indeed, it is correct and we will fix wording by meaning total molecular energies.
>
> **Q: “Dipoles and partial charges are trivial to obtain once DFT is converged”.**
>
> A: While obtaining dipoles/charges from one converged DFT calculation is cheap, large-scale ML datasets rarely store the wavefunctions or electron densities needed to recompute them. Many datasets only provide energies and forces. Regenerating partial charges for tens of thousands of geometries would require repeating all DFT calculations, which is orders of magnitude more expensive than using only the provided labels.
>
> **Q: “This is not convincing, if the Laplacian has a degenerate eigenspace the order will be arbitrary and specifics will be subject to numerical noise, a conjugated gradient solver doesnt change this”**
>
> A: We thank the reviewer for noting this inconsistency. LOBPCG returns a numerically consistent orthonormal basis for the span of the top-k eigenvectors, even when some eigenvalues are close or equal. Φ-Module does not rely on any ordering inside this subspace, so degeneracy does not affect its behavior. Additionally, in molecular graphs with continuously varying distance-based weights, exact degeneracy almost never occurs.

---

### Official Review · Reviewer_GKpD · 2025-11-02

**Soundness:** 3
**Presentation:** 3
**Contribution:** 2
**Rating:** 2
**Confidence:** 5

**Summary:**

The paper introduces Φ-Module, a physics-informed plugin designed to integrate electrostatic interactions into graph neural networks in a self-supervised way. By enforcing Poisson’s equation within the message-passing framework, Φ-Module enables models to learn atomic potentials and charges represented in the Laplacian eigenbasis of molecular graphs. This approach captures long-range electrostatic effects that standard local message passing often misses. The module includes a lightweight subnetwork, α-Net, that predicts eigenbasis coefficients, allowing derivation of an electrostatic energy term that improves total energy predictions with minimal computational overhead. Experiments on the OE62 and MD22 benchmarks show accuracy gains across several neural potentials

**Strengths:**

1. The paper proposes an integration of a first-principles physical law (Poisson’s equation) into GNN-based interatomic potentials. By embedding the Poisson constraint, the model learns to produce physically meaningful electrostatic potentials and charges in a self-supervised way, without requiring any ground-truth charges or external fields.
2. Φ-Module is designed as a universal augmentation that can be attached to essentially any GNN architecture for molecules. The authors demonstrate this generality by incorporating Φ-Module into multiple established models
3. A claimed advantage is that Φ-Module introduces very little overhead. The spectral α-Net and Laplacian eigenbasis computation are lightweight, adding roughly 5–10% to training time per epoch and a modest amount of memory usage.

**Weaknesses:**

1. The biggest weakness is generalizability and scalability. Experiments are restricted to OE62 and MD22. There’s no end-to-end training on truly realistic, large, diverse, million–sample datasets (e.g., OMol 25) or cross-dataset transfer demonstrating generalization across broader chemistries. As a result, it’s unclear how Φ-Module scales in sample size or generalizes to diverse systems.
2. Although the overall results favor Φ-Module, the improvements are not uniformly overwhelming. In OE62, one baseline GNN saw only ~5% error improvement with Φ-Module, which is relatively modest. On MD22, the Φ-augmented model did not win on every single metric – the original ViSNet still had the best outcome on 2 of the 14 comparisons, and in 3 of 14 cases Φ-Module failed to set a new state-of-the-art. This indicates that the benefits, while present, can be incremental, task-dependent, or just random (no error bar / std is given)
3. As a plug-in, Φ-Module adds four new hyperparameters and extra computations (solving for eigenvectors) to a model. While the authors argue this is lightweight, one might be concerned about implementation complexity.

**Questions:**

1. Could you clarify how the Laplacian eigenpairs are computed during training? The paper suggests using a fixed number k of eigenvectors and a batched eigendecomposition approach. Is this done via an iterative solver each message-passing step, or are eigenvalues computed once per epoch/structure and reused?
2. The paper references alternatives like adding Ewald summation to GNNs or using pre-computed partial charges. Did you consider comparing Φ-Module’s performance to such methods?
3. Beyond the tasks in this paper, how general is Φ-Module’s applicability? For instance, can it handle systems with periodic boundary conditions (common in materials simulations where Ewald is often needed)?

---

> ### Author Response · Authors · 2025-11-22
> **Rebuttal by Authors**
>
> We sincerely thank the reviewer for raising critical points. We address them below.
>
> **Q: There’s no end-to-end training on truly realistic, large, diverse, million–sample datasets (e.g., OMol 25) or cross-dataset transfer demonstrating generalization across broader chemistries & Although the overall results favor Φ-Module, the improvements are not uniformly overwhelming.**
>
> A: We are currently in the process of training models (SchNet and GemNet-O) on million-size large OMol25 dataset which will hopefully show how Φ-Module helps treating molecules with diverse long-range interaction. We will post results soon. We thank the reviewer for patience!
>
> **Q: While the authors argue this is lightweight, one might be concerned about implementation complexity.**
>
> A: We thank the reviewer for addressing complexity concerns. We attached the code in the supplementary materials with proper implementation which can be easily used with any NNIP. Additionally, we provide computational comparison in terms of speed and memory in Appendix C. Φ-Module outperforms other long-range method like Ewald summation or Neural-P3M in efficiency.
>
> **Q: Could you clarify how the Laplacian eigenpairs are computed during training?**
>
> A: Indeed, eigendecomposition is computed for each structure once in an epoch and then used to refine $\phi$ and $\rho$ at each message passing step. In practice, they can be computed once and stored for further efficiency. However, even without such design, our method performs favorably in terms of computational complexity (see Appendix C).
>
> **Q: The paper references alternatives like adding Ewald summation to GNNs or using pre-computed partial charges**
>
> A: We appreciate the reviewer for referencing alternatives. We compared our Φ-Module to Ewald summation in the main experiment on OE62 (see Figure 3 and Table 3). Φ-Module performs on-par with Ewald, but with a reasonably lower computational cost.
>
> **Q: Beyond the tasks in this paper, how general is Φ-Module’s applicability? For instance, can it handle systems with periodic boundary conditions (common in materials simulations where Ewald is often needed)?**
>
> A: We thank the reviewer for noting the importance of PBC conditions. Φ-Module can be easily extended to PBC scenario by adapting molecular graph to periodic cell. We note that Φ-Module doesn’t natively perform reciprocal-space sums, however it would function reliably given proper supercell definition. In our study we focused on regular diverse range of molecules without periodicity and molecular dynamics.

---

### Meta-Review · Area_Chair_B1fA · 2026-01-07

**Summary:**

The submission proposes a plug-in module for pretraining an interatomic potential neural network model. The idea is to enforce the model to produce a representation that is capable to support electrostatic potential and charge density prediction in a physically consistent way (i.e., satisfying the Poisson equation).

Reviewers acknowledge the physics-inspired innovation and the effect to capture long-range interaction, the ease of use in terms of base architecture choice and computational cost, and the empirical improvements.

Major concerns from the reviewers include:
1. Experiments do not show the potential to scale to large-scale pretraining datasets or any evidence of the trend. Empirical improvement is not consistently significant.
2. The $\Phi$-module may seem limited to non-periodic systems and long-range effects of only electrostatic interactions.
3. Reviewer gZro pointed out a few improper statements in the paper.
4. The rationale of the proposed method requires further justifications (effect of the predicted potential and charge, weights for the Laplacian, potential risk of discontinuity around the radius cutoff threshold, violation of permutation invariance, effect of the PDE loss).
5. The approach is not demonstrated on more modern architectures.

**Reviewer Concerns:**

After reading authors' rebuttal, I would treat concerns 2,3,5 as minor issues, though they are flaws that degrade quality.

I would still treat other concerns as reasonable challenges to the paper's major contribution.

1\. I agree with Reviewer QJiW that to demonstrate the claimed benefit of capturing long-range interaction, experimental demonstrations on larger molecules are expected.

4\. I would take the general interpretation acceptable but with a degraded physical meaning: the $\Phi$-module pretraining mainly mimic the computational pattern for the Coulomb interaction instead of exactly the electrostatic energy, which releases my critics on the effect of the predicted potential and charge and the weights for the Laplacian. Nevertheless, potential risk of discontinuity around the radius cutoff threshold and violation of permutation invariance are still the downsides. Moreover, I hold a similar confusion as Reviewer oDat: if the two alpha-nets learn the same output, then would the pretraining target be perfectly satisfied?

5\. This is also an insufficiency that makes the results less convincing.

**Reviewer Scores:**

I would not speculate that the negatively rating reviewers would increase their scores.

---

### Decision · Program_Chairs · 2026-01-26

Reject